# Can We Successfully Discontinue Anti-Tumor Necrosis Factor-α Treatment in Children with Non-Systemic Juvenile Idiopathic Arthritis? The Experience of a Tertiary Center

**DOI:** 10.3390/biomedicines13102329

**Published:** 2025-09-24

**Authors:** Ekaterina I. Alexeeva, Irina T. Tsulukiya, Tatyana M. Dvoryakovskaya, Dmitry A. Kudlay, Ivan A. Kriulin, Maria S. Botova, Natalya M. Kondratyeva, Elizaveta A. Krekhova, Meiri Sh. Shingarova, Maria Y. Kokina, Anna N. Fetisova, Kseniya B. Isaeva, Aleksandra M. Chomakhidze, Christina V. Chibisova, Mikhail M. Kostik

**Affiliations:** 1Department of Pediatric Rheumatology, National Medical Research Center of Children’s Health, Moscow 119991, Russia; alekatya@yandex.ru (E.I.A.); irinatsulukiya@gmail.com (I.T.T.); tbzarova@mail.ru (T.M.D.); 79671819676@yandex.ru (I.A.K.); mariabotova22@gmail.com (M.S.B.); 131nk@mail.ru (N.M.K.); lizakrek@mail.ru (E.A.K.); mshingarova@mail.ru (M.S.S.); kokinamariah@yandex.ru (M.Y.K.); anna_534@mail.ru (A.N.F.); isaeva-kseniya@yandex.ru (K.B.I.); achomakhidze@mail.ru (A.M.C.); chibisova.kri@yandex.ru (C.V.C.); 2Clinical Institute of Children’s Health Named After N.F. Filatov, Department of Pediatrics and Pediatric Rheumatology, I.M. Sechenov First Moscow State Medical University (Sechenov University), Moscow 119435, Russia; 3Association of Pediatric Rheumatologists, Moscow 107078, Russia; 4Department of Pharmacology, Institute of Pharmacy, I.M. Sechenov First Moscow State Medical University (Sechenov University), Moscow 119435, Russia; d624254@gmail.com; 5Laboratory of Personalized Medicine and Molecular Immunology, National Research Center—Institute of Immunology Federal Medical-Biological Agency of Russia, Moscow 115522, Russia; 6Hospital Pediatry, Saint-Petersburg State Pediatric Medical University, Saint-Petersburg 194100, Russia

**Keywords:** juvenile idiopathic arthritis, biologic, etanercept, medicine tapering, clinical remission, outcomes, anti-TNF-α, disease activity

## Abstract

**Background:** Some patients with juvenile idiopathic arthritis (JIA) can successfully undergo withdrawal of treatment with anti-tumor necrosis factor alpha (anti-TNF-α) therapy, which may reduce economic and treatment-related burdens and the potential morbidity of treatment for at least 6 months. Currently, no guidelines exist on the appropriate withdrawal of anti-TNF-α therapy once clinically inactive disease (CID) has been achieved. This study aimed to assess the possibility of withdrawing anti-TNF-α therapy in children with non-systemic JIA after achieving long-term clinical remission. **Methods:** This single-center retrospective cohort study included data from 137 non-systemic JIA patients treated with anti-TNF-α therapy (etanercept or adalimumab) and having maintained CID for at least 24 months during treatment. Demographic, laboratory, and treatment data were collected at JIA onset, at the initiation of anti-TNF-α therapy, every 6 months during therapy, and at the time of disease flare. Anti-TNF-α therapy was discontinued abruptly after discussing it with the patients and their families in each case. Outcomes were assessed using standard criteria for remission in JIA (ACRpedi and Wallace criteria). **Results:** Following withdrawal of TNF-α inhibitors, 93/137 patients (67.9%) experienced a disease flare, with a median time to flare of 7 months (3; 14). Thirty-two percent of patients remained in remission for a median of 63 months. Absence of flare during the first 22 months after discontinuation was associated with prolonged biologic-free remission (odds ratio 682; 95% CI 38.6–12,062; *p* < 0.0001), with 83% sensitivity and 100% specificity (area under the ROC curve 0.967). Most flares involved arthritis (76%) and/or uveitis (24%), primarily affecting knees, ankles, and wrists. Inflammatory markers were generally lower at flare compared to baseline. Biological therapy was resumed in 84/93 patients (90.3%), achieving at least a 50% improvement according to ACRpedi criteria within 3 months and remission according to Wallace criteria within 6 months. **Conclusion:** Over two-thirds of patients with non-systemic JIA who achieve CID experience a flare within seven months of anti-TNF-α discontinuation. Re-initiation of biologic therapy is effective in restoring remission. These results indicate that prolonged biologic-free remission is possible in a subset of patients, highlighting the need for individualized withdrawal strategies and careful post-discontinuation monitoring.

## 1. Introduction

Juvenile idiopathic arthritis (JIA) is the most common rheumatic disease in children. It is defined as inflammatory arthritis of unknown etiology occurring in individuals under 16 years of age, present for more than 6 weeks, and diagnosed by exclusion [1]. JIA is classified into seven subsets according to the International League of Associations for Rheumatology (ILAR): oligoarticular, polyarticular (RF-positive and RF-negative), systemic, enthesitis-related arthritis, psoriatic arthritis, and undifferentiated arthritis [2].

Recent advances in treatment, including anti-tumor necrosis factor alpha (anti-TNF-α) therapy, have significantly improved outcomes for children with JIA [3]. Inactive disease and disease remission have become attainable goals, and biologics have played an important role in achieving them. The potential risks of immunosuppression include infections, particularly mycobacterial and fungal infections, as well as the possibility of developing secondary malignancies with TNF blockers [4]. In JIA, the incidence of serious infections during anti-TNF-α therapy ranges from approximately 2 to 5 per 100 patient-years, while the risk of malignancy appears to be very low, with large registries not showing a clear increase compared to the general pediatric population. Although uncommon, these risks are clinically significant and contribute to the rationale for considering treatment withdrawal. Minimizing the use of anti-TNF-α could lead to decreased adverse effects and costs [5].

The long-term use of biological drugs to maintain remission in JIA is associated with a significant economic burden for the healthcare system [6]. IA affects approximately 16–150 children per 100,000 population worldwide, and up to 20–30% of patients in developed countries receive biologic therapy. The annual cost of long-term anti-TNF-α therapy in pediatric patients is typically estimated at USD 10,000–20,000 per patient. The non-economic burden of JIA therapy and the so-called “hidden costs”—the psychological burden of frequent injections and “lifelong” drug therapy—are also significant [7].

Although formal, widely adopted guidelines for anti-TNF-α discontinuation in pediatric JIA are lacking, some pediatric rheumatology working groups have suggested tapering approaches or structured monitoring after biologic cessation; however, these recommendations are heterogeneous and not standardized in terms of timing, patient selection, or follow-up protocols [8].

Because systemic JIA has a distinct pathogenesis, clinical course, and treatment strategy, we focused this study on non-systemic JIA, where withdrawal decisions are more clinically relevant and evidence is particularly scarce. Several studies in adults with rheumatoid arthritis have explored factors associated with the successful discontinuation of anti-TNF-α therapy, but pediatric data remain limited [9]. Small cohort and pilot studies in children with JIA suggest that a longer duration of clinically inactive disease before withdrawal, younger age at disease onset, and absence of concomitant methotrexate may increase the likelihood of maintaining remission; however, findings are inconsistent, and no robust predictors have been universally confirmed.

The aim of this study was to evaluate the feasibility and outcomes of anti-TNF-α discontinuation in children with non-systemic JIA who achieved long-term clinical remission and to explore potential predictors to inform individualized withdrawal strategies.

## 2. Patients and Methods

### 2.1. Study Design and Setting

This single-center retrospective cohort study included all patients with JIA treated with TNF-α inhibitors after methotrexate failure at the National Medical Research Center of Children’s Health, Moscow, Russian Federation, Division of Pediatric Rheumatology, between 1 December 2004, and 31 August 2018.

### 2.2. Participants

Methotrexate failure was defined as inadequate clinical response after at least 3 months of therapy at the recommended dose, or intolerance preventing continuation of treatment.

Patients who received etanercept 0.8 mg/kg/week or adalimumab 24 mg/m^2^ every other week and whose disease remained inactive for 24 months were considered candidates for discontinuation of TNF-α inhibitors.

Anti-TNF-α therapy was discontinued abruptly, which represented the standard practice at our center for eligible patients, following discussion with parents and patients. Tapering was not routinely performed. The majority of patients continued non-biologic DMARD therapy, including methotrexate (10–15 mg/m^2^/week), leflunomide (0.2 mg/kg/day), or sulfasalazine (20–30 mg/kg/day), according to standard pediatric dosing.

The inclusion criteria were the following: (i) fulfillment of the ILAR criteria for JIA; (ii) age ≤ 18 years at enrollment; (iii) non-systemic JIA subtypes only (oligoarticular, polyarticular RF-negative, enthesitis-related); (iv) treatment with anti-TNF-α therapy (etanercept or adalimumab) during the study period; (v) disease duration ≥ 6 months at enrollment; and vi) remission ≥ 24 consecutive months on biologics. The exclusion criterion was limited follow-up or observation deemed inadequate for extracting reliable data, defined as fewer than 3 clinic visits or follow-up of less than 12 months. Figure 1 illustrates patient selection and grouping.

### 2.3. Definitions and Outcomes

For this study, non-biologic remission was defined as inactive disease for ≥12 months off anti-TNF-α therapy according to the C. Wallace criteria [10]. Other immunosuppressive therapies, excluding all kinds of corticosteroids and biologic agents, were permitted.

Flare was defined as no longer fulfilling inactive disease criteria, including the presence of active arthritis on a joint count, an increase in physician or parent global assessment (VAS), or elevated inflammatory markers (ESR or CRP), consistent with the Wallace criteria. Transient synovitis episodes that resolved by the next visit and required only NSAIDs were not considered flares. The primary outcome was the proportion of patients achieving non-biological remission more than 12 months after anti-TNF-α discontinuation.

### 2.4. Data Collection

Electronic medical records were used. Data were anonymized prior to analysis, ensuring assessors were blinded to patient identity and outcomes. Laboratory evaluations included ESR, CRP, ANA, HLA B27, and other routine parameters. Demographic, clinical, and treatment data were collected at disease onset, at the initiation of anti-TNF-α therapy, every 6 months during therapy, and at flare.

### 2.5. Statistics Analysis

This was a retrospective, observational cohort study. No formal sample size calculation was conducted due to the retrospective design. The statistical analysis was primarily descriptive. Quantitative variables were assessed for normality using the Shapiro–Wilk test; a *p*-value > 0.05 indicated a normal distribution. Continuous data were expressed as medians and interquartile ranges (IQRs), while categorical variables were summarized as absolute numbers and percentages.

Comparisons of categorical variables between groups (e.g., patients with and without disease flare) were performed using Pearson’s chi-square test. For continuous variables not normally distributed, the Wilcoxon signed-rank test was used for paired comparisons. Statistical significance was defined as a two-tailed *p*-value < 0.05.

Logistic regression analysis was used to estimate the odds ratio (OR) and 95% confidence intervals (CIs) for the association between absence of flare within the first 22 months after anti-TNF-α discontinuation and prolonged biologic-free remission.

Kaplan–Meier survival curves were plotted to depict the time to disease flare after anti-TNF-α discontinuation, providing a visual representation of the proportion of patients maintaining non-biologic remission over the follow-up period. Patients who did not experience a flare were censored at the time of their last follow-up.

The predictive accuracy of “time to flare” as a marker for sustained biologic-free remission was evaluated using receiver operating characteristic (ROC) curve analysis, with the sensitivity, specificity, and area under the curve (AUC) reported.

Statistical analysis and visualization of the data were performed using R version 4.2.2 (R Foundation for Statistical Computing, Vienna, Austria).

### 2.6. Ethics

This study was conducted according to the Declaration of Helsinki. The Ethics Committee of the National Medical Research Center of Children’s Health approved the study protocol (No.12, 6 October 2020). Written informed consent was obtained from all patients or their legal guardians (for patients aged < 15 years). All patient data were anonymized to protect confidentiality.

## 3. Results

A total of 959 patients treated with anti-TNF-α therapy were screened. The duration of remission was ≥24 consecutive months for 179 patients, and 148 patients discontinued anti-TNF-α therapy. For the subsequent analysis, eligible data from 137 patients (who completed the study) were included, and 11 patients were excluded due to incomplete medical records (the patient’s care was transferred to another site without follow-up). All patients were divided into two groups, depending on the time of flare: (1) flare within 22 months; (2) flare after 22 months or maintained non-biologic remission at the last available follow-up visit (Figure 1).

### 3.1. Demographic Characteristics

One hundred and thirty-seven patients (37/137, 27% male; 100/137, 73% female) with a median age at disease onset of 3 years (IQR 1.8–5.9 years) were included in the analysis. The IA subtypes were oligoarticular 76/137 (55%) [persistent 58/137 (42%) and extended 18/137 (13%)], polyarticular RF-negative 41/137 (30%), and enthesitis-related arthritis 20/137 (15%). No patients with RF-positive polyarthritis achieved 24-month remission.

Thirty-five patients (26%) had a history of uveitis; HLA–B27 positivity was observed in 23/137 (17%) and ANA positivity in 15/137 (11%).

Sixty (43.8%) patients received non-biological anti-rheumatic drugs—methotrexate 55/60 (92%), sulfasalazine 3/60 (5%), a combination of methotrexate with sulfasalazine 1/60 (2%), and methotrexate with cyclosporine 1/60 (2%)—at the time of anti-TNF-α-α withdrawal. During the observation period when anti-TNF-α therapy was stopped, non-biologic DMARDs were discontinued in 14/60 patients (23.3%) due to adverse events.

A total of 116/137 patients (85%) were biologic-naïve, and 21/137 (15%) had received another (non-anti-TNF-α) biologic drug prior to study inclusion. They were switched to anti-TNF-α therapy due to the failure of their initial biological treatment. The majority of patients (95/137, 69%) received etanercept, followed by adalimumab (42/137, 31%). At anti-TNF-α discontinuation, 57 patients (42%) continued methotrexate therapy. The association between methotrexate continuation and risk of flare was not analyzed in this study.

The median age at initiation of anti-TNF-α therapy was 6.1 years (IQR 3.5–9.9), and the median disease duration prior to starting anti-TNF-α therapy was 19 months (IQR 9–45). The median time from disease onset to initiation of TNF-α therapy was 19 months (IQR 9–45), with a median duration of therapy of 42 months (IQR 36–54) and median remission on therapy of 35 months (IQR 29–46).

Demographics, disease characteristics, and treatment details are presented in Table 1.

### 3.2. Withdraw of TNF-α Inhibitors

The vast majority of enrolled patients (93/137, 67%) experienced flares after stopping treatment with biologics. The median time to flare was 7 months (IQR 3–14) [min. 1; max. 58], and no significant differences were found between the types of arthritis. The remaining 44 patients (32%) did not develop a disease flare and remained in long-term biologic-off remission for a median of 63 months (42; 82) [min. 24; max. 119] before the last available follow-up visit (Figure 2).

Figure 3 illustrates the predictive performance of “time to flare” (<22 months) in distinguishing between patients with and without flare events. In the left panel, a scatter plot shows the distribution of “time to flare” values in patients with and without flares. The optimal cut-off time identified for prediction was 22 months, achieving a sensitivity of 82.8% and a specificity of 100%.

The right panel displays the receiver operating characteristic (ROC) curve for the model, demonstrating significant discriminative ability with an area under the curve (AUC) of 0.967 (95% CI: 0.921–0.990). The curve shows high sensitivity across a wide range of specificity values, confirming the robustness of “time to flare” as a predictive marker.

The absence of a flare during the first 22 months after anti-TNF-α discontinuation was significantly associated with prolonged biologic-free remission. Among patients who remained flare-free for at least 22 months, 44/44 (100%) maintained sustained remission. In contrast, none of the 93 patients who flared earlier achieved long-term remission (OR ≈ 682, 95% CI: 38.6–12,062, *p* < 0.0001), suggesting a significant association. However, the wide confidence interval indicates variability due to the zero-event group.

### 3.3. Characteristics of Flares

The flares were characterized by relapse of arthritis alone in 71/93 (76.3%) patients, uveitis alone in 11/93 (11.8%), and both arthritis and uveitis in 11/93 (11.8%). The affected joints during flare included the knee (72/82, 88%), ankle (25/82, 30.5%), wrist (6/82, 7%), small joints of the hands (6/82, 7%) and feet (3/82, 4%), hip joints (1/82, 1.5%), and sacroiliac joints (1/82, 1.5%). The median number of joints with active arthritis per patient at flare was two (IQR, 1–6), significantly lower than that at disease onset (median, 4; IQR, 2–7; *p* < 0.0001, Wilcoxon signed-rank test).

In two patients (2%), TNF-α inhibitors were withdrawn due to the development of de novo uveitis.

Among patients with uveitis, flare-ups occurred in 8/22 (36%) within 6 months, 3/22 (14%) within 12 months, 6/22 (27%) within 18 months, and 5/22 (23%) beyond 18 months.

At the time of the JIA flare after anti-TNF-α discontinuation, laboratory markers of inflammation were generally low. The median erythrocyte sedimentation rate (ESR) was 6 mm/h (IQR 3–15), and the median serum concentration of C-reactive protein (CRP) was 2 mg/L (IQR 2–5). An elevated ESR (>20 mm/h) was observed in 14/93 patients (15%) and elevated CRP (>5 mg/L) in 24/93 patients (26%). In contrast, at the time of disease onset, the inflammatory markers were substantially higher. The median ESR was 20 mm/h (IQR 12–38), and the median CRP was 5 mg/L (IQR 2–20), with some patients demonstrating values as high as 100 mm/h for the ESR and 177 mg/L for CRP.

In 72/93 (77.4%) patients with arthritis flare, the joint involvement pattern was similar to that observed prior to TNFα inhibitor discontinuation, affecting joints previously involved; in 21/93 (22.6%) patients, at least one newly affected joint was observed at flare.

### 3.4. Efficacy of Restarting TNF-α Inhibitor Therapy

Of the 137 patients for whom anti-TNF-α therapy was withdrawn due to disease control, 84 (61%) were restarted on therapy by the local pediatric rheumatology team following a flare, as documented at the last recorded follow-up.

Among the 84 patients who restarted therapy due to flare, 70 (83%) resumed the same biologic, 7 (8%) switched to a different biologic, and 7 (8%) required only NSAIDs or methotrexate for mild symptoms.

Clinically inactive disease, according to the C. Wallace criteria, was achieved in all patients thereafter.

## 4. Discussion

This study provides valuable insights into the outcomes of anti-TNF-α therapy withdrawal in a large cohort of children with juvenile idiopathic arthritis (JIA). Our findings show that a significant proportion of patients (67%) experienced disease flare-ups after discontinuation of anti-TNF-α therapy. These data align with those from prior international studies, which reported flare rates ranging from 60% to 80% [11,12]. Notably, the median time to flare was 7 months, indicating that the highest risk of disease reactivation occurs within the first year after therapy cessation, consistent with earlier research.

The biological and immunological mechanisms underlying flare risk or prolonged remission after anti-TNF-α discontinuation remain unclear. Factors such as immunological memory, the synovial environment, and the strategy of abrupt versus tapered discontinuation may play a role. Although laboratory data including ANA, HLA B27, ESR, and CRP were collected, we did not analyze their potential predictive value for flare or sustained remission. Future investigations should include biomarker-driven analyses and mechanistic studies to clarify predictors of long-term remission.

Although reintroduction of TNF-α inhibitors was highly effective in our cohort, the health and economic implications of attempting treatment withdrawal should be considered. Given that approximately 67% of patients experienced disease flare, the cost-effectiveness of withdrawal trials may be limited, particularly when factoring in the clinical monitoring required and the morbidity associated with flares. This is particularly relevant in healthcare systems where biologic costs and monitoring resources represent a substantial burden. Balancing the potential benefits of reduced drug exposure, such as lower risks of adverse effects and decreased treatment costs, against the risk and burden of flare is essential. These findings also underscore the need for evidence-based guidelines for biologic withdrawal, as no formal protocols currently exist. Standardized, evidence-based algorithms for tapering or cessation are urgently needed.

We identified the cut-off “time to flare” as a potent predictor of prolonged biologic-free remission. The absence of flare within the first 22 months after biologic withdrawal was associated with prolonged biologic-free remission (AUC 0.967), indicating high discriminative ability in this cohort. While the absence of flare within the first 22 months was significantly predicted of prolonged biologic-free remission, the very wide confidence interval (OR ≈ 682, 95% CI 38.6–12,062) reflects statistical instability due to a zero-event group. This limitation should be acknowledged, and the predictive cut-off should be interpreted with caution. The 22-month cut-off for absence of flare may serve as a clinically useful marker to guide decisions on biologic withdrawal and post-withdrawal monitoring. In practice, this could help identify patients requiring closer surveillance during the first two years and support shared decision-making with families. Further prospective studies are needed to validate and refine such algorithms for routine clinical practice. In comparison, similar studies reported AUC values ranging from 0.80 to 0.90, suggesting that the discriminative ability observed in our cohort is within or above this range [13,14].

Comparing our results with those of Simonini et al. [8], we observed similarities. Both studies reported a flare rate of approximately 67% after cessation of biologic therapy, with around 30% of patients remaining in remission long-term. However, Simonini et al. noted that 15% of patients experienced uveitis flare-ups, while our study observed a slightly lower level (11.8%). The slightly lower incidence of uveitis flares in our cohort may reflect differences in patient selection, treatment strategies, or definitions of uveitis flare.

Comparing our findings with those of Chang Y. et al. [15], which demonstrated successful discontinuation of TNF-α inhibitors, our results support the observation of sustained remission following treatment cessation. In the study conducted by Su et al., the remission lasted an average of 26 months. In prospective, open-label study by Y. Cai et al. [16], a reduction in etanercept dosage resulted in a low flare rate (12%) within the first 12 months, with no flares in the subsequent year.

Our study also corroborates the findings of K. Baszis et al. [17], who reported that 32% of patients maintained remission without biologic therapy. In their study, 25% of patients experienced flare-ups within 3 months of stopping TNF-α inhibitors.

In line with a study by N.R. Acharya et al. [18], our findings underscore the risk of flare-ups, particularly uveitis, following discontinuation of anti-rheumatic therapies. Acharya et al. found that uveitis flare-ups occurred an average of 2.8 months after corticosteroid withdrawal and 16.7 months after anti-TNF-α therapy cessation. Our data show that 36% of patients experienced uveitis flare-ups within six months of stopping biologic therapy, and 23% had flare-ups more than 18 months after stopping biologic therapy.

Moreover, M.A. Lerman et al. [19] confirmed that discontinuation of biologic therapy, especially TNF-α inhibitors, increases the likelihood of relapse. This supports our findings, which show that a significant proportion of patients experienced disease flare-ups after stopping biologic therapy. Taken together, these results highlight the importance of long-term vigilance and individualized risk assessment.

Our results align with findings from recent meta-analyses and international guidelines, which report flare rates after anti-TNF-α discontinuation ranging from 60% to 80% and underscore the importance of individualized tapering strategies and careful monitoring [20,21]. By providing data from a larger single-center cohort with extended follow-up, our study complements these analyses and offers additional insight into the predictive value of early absence of flare for prolonged biologic-free remission.

The reintroduction of anti-TNF-α inhibitors in patients who relapsed was highly effective, with all patients achieving clinically inactive disease according to the Wallace criteria. Our data are consistent with those from prior studies [3,22], supporting the effectiveness of re-treatment with anti-TNF-α inhibitors even after a period of discontinuation. Furthermore, the majority of patients were able to resume the same biologic agent without the need for switching, suggesting that the drug’s efficacy remains intact despite treatment interruptions.

Recent studies suggest that gradual tapering of anti-TNF therapy may reduce flare risk compared with abrupt discontinuation in children with JIA [23]. Certain inflammatory biomarkers, such as IL-1β and S100A9, have been shown to predict long-term remission and flare risk in pediatric JIA [24]. Long-term follow-up indicates that a substantial proportion of children maintain biologic-free remission, although factors such as ANA positivity and disease duration influence relapse risk [25].

Future directions should include prospective controlled trials comparing abrupt versus tapered discontinuation of anti-TNF-α therapy, studies integrating biomarkers and advanced imaging modalities such as ultrasound or MRI, and the use of machine learning prediction models to identify patients most likely to maintain remission. Additionally, cost-effectiveness modeling could inform clinical decision-making and guide policy on biologic withdrawal in pediatric JIA. These strategies will be crucial for developing standardized, evidence-based protocols that optimize both clinical outcomes and healthcare resource utilization.

## 5. Limitations

This study has several limitations. First, the retrospective design may have introduced both selection and information bias: patients who maintained long-term remission were more likely to be included, and some flare assessments depended on documentation that may have been incomplete or subjective. Second, the heterogeneity of JIA subtypes and treatment regimens complicates interpretation, as oligoarticular versus polyarticular forms, or ANA- and HLA B27-positive patients, may carry different risks of flare that could not be formally analyzed. Third, the single-center setting and relatively limited sample size may reduce the generalizability of the findings to broader populations. Fourth, the very high odds ratio observed for the absence of flare within the first 22 months reflects statistical instability due to a zero-event group and should therefore be interpreted with caution. Fifth, incomplete long-term follow-up in some patients may have led to underestimation of late relapses or complications.

Furthermore, our study assessed only abrupt discontinuation of anti-TNF-α therapy, without comparison to tapering strategies, which are increasingly favored in clinical practice. This limits the applicability of our findings to tapering contexts. Additionally, many patients continued non-biologic DMARDs during anti-TNF-α withdrawal; the influence of concomitant therapies on flare risk could not be disentangled, introducing potential confounding. Finally, the observational nature of the study does not allow causal inferences. Consequently, these results should be considered hypothesis-generating rather than confirmatory, and future prospective, multi-center studies with stratification by JIA subtype, biomarker integration, and standardized tapering protocols will be needed to validate and extend these findings.

## 6. Conclusions

This study provides valuable insights into the long-term outcomes of discontinuing anti-TNF-α therapy in children with non-systemic JIA. While a meaningful proportion of patients (32%) achieved prolonged biologic-free remission, the majority (67%) experienced disease flare, typically around 7 months after cessation. Uveitis emerged as a frequent and clinically important cause of flare, particularly within the first year, underscoring the need for vigilant monitoring.

The absence of flare within the first 22 months appeared to be a significant predictor of sustained remission, suggesting that the early post-withdrawal disease course may help guide long-term management. Reintroduction of biologic therapy after relapse was generally effective in restoring remission, highlighting the importance of timely intervention to prevent prolonged uncontrolled inflammation.

These findings emphasize the need for individualized withdrawal strategies, considering patient-specific factors such as JIA subtype and the presence of uveitis. The influence of concomitant non-biologic DMARDs must also be acknowledged when interpreting remission rates, as they may contribute to maintenance of disease control.

From a clinical and policy perspective, our results reinforce the necessity of developing evidence-based guidelines for biologic withdrawal, balancing reduced drug exposure with the risks of relapse. Future research should focus on integrating clinical, laboratory, and imaging biomarkers; evaluating tapering protocols versus abrupt discontinuation; and assessing cost-effectiveness to optimize biologic-free remission strategies.

## Figures and Tables

**Figure 1 biomedicines-13-02329-f001:**
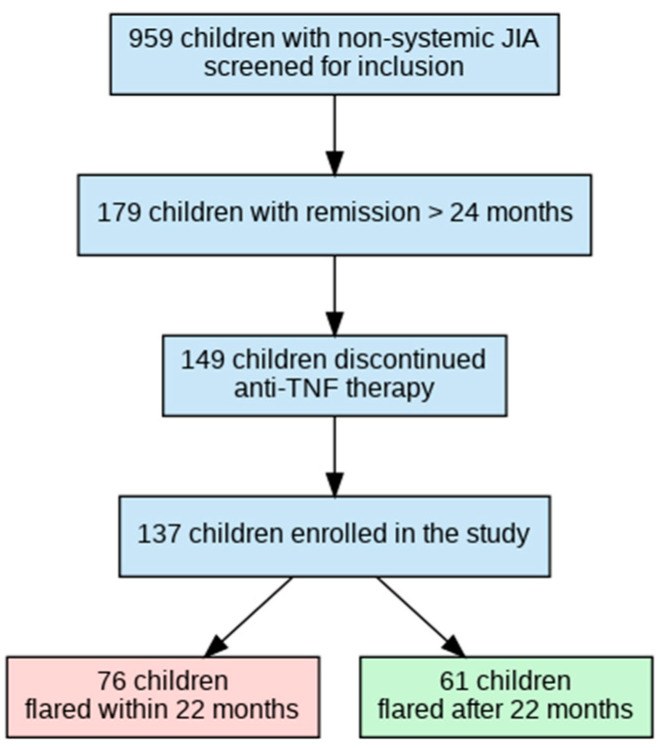
Flowchart of patient selection and grouping by time to flare after anti-TNF-α withdrawal.

**Figure 2 biomedicines-13-02329-f002:**
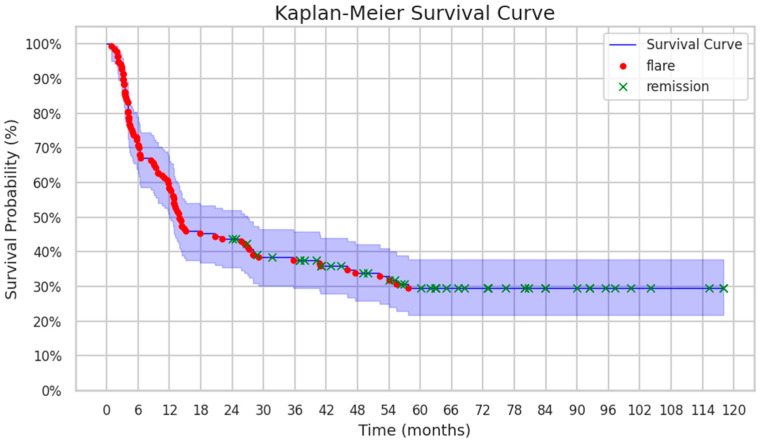
Kaplan–Meier survival curve demonstrating flare-free survival (FFS) of patients after discontinuation of anti-TNF-α therapy.

**Figure 3 biomedicines-13-02329-f003:**
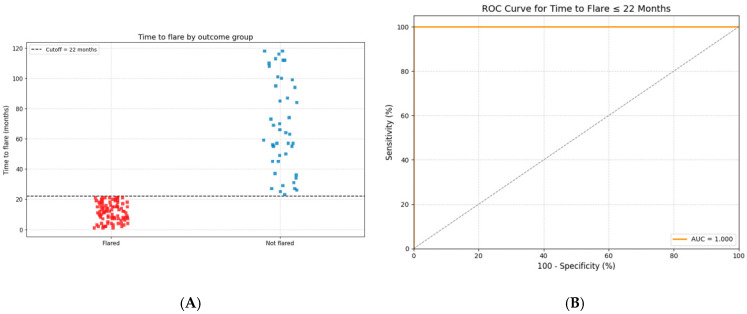
Time to flare as a predictor of prolonged biologic-free remission after anti-TNF-α withdrawal. (**A**) The distribution of “time to flare” among patients who experienced a flare (Flared) versus those who remained in sustained remission (Not flared). Each square represents an individual patient. The dashed horizontal line marks the optimal cut-off at 22 months. Patients with a time to flare of more than 22 months were significantly more likely to maintain long-term remission. (**B**) A receiver operating characteristic (ROC) curve assessing the predictive accuracy of “time to flare ≤ 22 months” for identifying patients at risk of flare. The model demonstrated significant discriminative performance, with an area under the curve (AUC) of 0.967 (95% CI: 0.921–0.990), a sensitivity of 82.8%, and a specificity of 100%.

**Table 1 biomedicines-13-02329-t001:** Characteristics of the 137 JIA patients being in remission or flared after anti-TNF-α therapy was withdrawn.

Characteristics	Total, *n* = 137	Flared, *n* = 76	In Remission, *n* = 61
Gender, *n* (%)			
• male	37 (27)	15 (20)	22 (36.1)
• female	100 (73)	61 (80)	39 (63.9)
Age at disease onset, Me (25%; 75%)	3 (1.8–5.9)	2.7 (1.7; 5.2)	4 (2; 6)
Age at start of anti-TNF-α, years, Me (25%; 75%)	6.1 (3.5; 9.9)	6.1 (3.5; 9.4)	6 (3.5; 10)
JIA category, *n* (%)			
• persistent oligoarticular	58 (42.3)	32 (42.1)	26 (42.6)
• extended oligoarticular	18 (13.1)	13 (17.1)	5 (8.2)
• RF-negative polyarticular	41 (29.9)	23 (30.3)	18 (29.5)
• enthesitis-related	20 (14.6)	8 (10.5)	12 (19.7)
Duration of disease prior to TNF-α inhibitors, months, Me (25%; 75%)	19 (9; 45)	25(11; 45)	14 (5; 36)
Active joints *, Me (25%; 75%)	4 (2; 7)	4 (2; 6)	3 (2; 7)
Swollen joints *, Me (25%; 75%)	3 (2; 6)	3 (2; 6)	3 (2; 6)
Painful joints *, Me (25%; 75%)	4 (2; 6)	4 (2; 6)	3 (2; 6)
Joints with limited motion *, Me (25%; 75%)	4 (2; 6)	4 (2; 6)	3 (2; 6)
Morning stiffness, min *, Me (25%; 75%)	30 (0; 60)	25 (0; 60)	40 (0; 35)
Active uveitis *, *n* (%)	35 (25.5)	22 (29)	13 (71)
ESR *, mm/h, Me (25%; 75%)	14 (8; 26)	14 (9.5; 26)	15 (8; 20)
CRP *, mg/L, Me (25%; 75%)	5 (2; 12)	5 (2; 12)	5 (3; 12)
ANA positivity, *n* (%)	15 (10.9)	8 (10.5)	7 (11.5)
HLA B27 positive, *n* (%)	23 (16.8)	13 (17.1)	10 (16.4)
MDVAS *, mm, Me (25%; 75%)	85 (84; 92)	85 (84; 88)	75 (70; 75)
Concomitant treatment *, *n* (%)	100 (73)	58 (76)	42 (68.9)
• methotrexate	88 (88)	55 (94.8)	33 (54.1)
• cyclosporine A	2 (2)	1 (2)	1 (1.6)
• sulphasalazine	5 (5)	2 (3.5)	3 (4.9)
• methotrexate + cyclosporine A	5 (5)	5 (8.6)	0
Biologic “naïve” patients, no. (%)	116 (84.7)	62 (81.6)	54 (88.5)
Time before the first biologics, months, Me (25%; 75%)	18 (12; 35)	18 (12; 35)	17 (14; 34)
Previous biologic therapy, no. (%)	21 (15.3)	15 (19.7)	6 (9.8)
• infliximab	19 (13.8)	14 (93)	5 (8.2)
• tocilizumab	1 (0.7)	1 (7)	0
• abatacept	1 (0.7)	0	1 (1.6)
Anti-TNF-α-a treatment, *n* (%)			
• etanercept	95 (69)	55 (72.4)	40 (65.6)
• adalimumab	42 (31)	21 (27.6)	21 (34.4)
Concomitant treatment **, no. (%)	60 (43.8)	22 (28.9)	38 (62.3)
• methotrexate	55 (31.7)	18 (81.8)	37 (60.7)
• sulphasalazine	3 (5)	2 (9.1)	1 (1.6)
• methotrexate + cyclosporine A	1 (1.7)	1 (4.5)	0
• methotrexate + sulphasalazine	1 (1.7)	1 (4.5)	0

* At the time of TNF-α antagonist start. ** At the time of TNF antagonist prescription. ANA—antinuclear antibody test, TNF—tumor necrosis factor, RF—rheumatoid factor, ESR—erythrocyte sedimentation rate, CRP—C-reactive protein, JIA—juvenile idiopathic arthritis, HLA B27—human leukocyte antigen B27.

## Data Availability

The data presented in this study are available on request from the corresponding author.

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
