# Peer review of "Can We Successfully Discontinue Anti-Tumor Necrosis Factor-α Treatment in Children with Non-Systemic Juvenile Idiopathic Arthritis? The Experience of a Tertiary Center"

_biomedicines, 2025, doi:10.3390/biomedicines13102329_

Round 1
Reviewer 1 Report
Comments and Suggestions for Authors
This is an important and clinically relevant study with strong data. But here are some comments to make it better:
In the abstract:
- I suggest to replace “spared the cost” with “reduce economic and treatment-related burden”.
- The authors need to specify whether this was single or multi-center study.
- It will be better to add a sentence on the clinical significance to the conclusion section mentioning for example that “These results suggest that prolonged biologic-free remission is possible in a subset of patients, highlighting the need for individualized treatment withdrawal strategies.”
In the introduction:
- The authors need to add the reason for focusing on non-systemic JIA earlier in the introduction.
- The authors mentioned that there are risks for anti-TNF treatment or its discontinuation but does not explain whether these risks are frequent, rare, or primarily theoretical in JIA. Adding incidence data (if available) would strengthen the justification.
- It will be better if the authors highlighted whether previous studies (small cohorts or pilot studies) have suggested predictors for successful withdrawal, as this would position the study within the existing evidence base.
- I suggest that the authors could reorganize introduction into four clear parts as follows: definition and burden of JIA, advances in treatment and role of anti-TNF, risks and burdens of long-term biologic use (medical, economic, psychological), and finally the gap in evidence and aim of current study. They also need to add more quantitative context as the prevalence of JIA, proportion treated with biologics, typical costs of long-term anti-TNF in pediatrics.
In the patients and methods:
- The timeframe is incomplete “between December 1, 2004 …” with no end date is given. This must be specified.
- The authors need to state whether this was a single center or multi-center study.
- The authors need to clarify whether ethics approval and informed consent were obtained, as this is mandatory for retrospective studies involving children.
- The inclusion criteria should specify age ranges, JIA subtypes included (why non-systemic only?), and disease duration at enrollment.
- Exclusion criteria are vague “limited follow-up or observation deemed inadequate” is subjective. Define this operationally (e.g., less than n visits, less than n months follow-up).
- The authors need to clarify whether methotrexate failure was defined objectively (e.g., inadequate response after ≥3 months, intolerance).
- Explain why discontinuation was abrupt rather than tapered, and whether this was standard practice at the center.
- Mention whether other DMARDs (methotrexate, leflunomide, sulfasalazine) were continued and at what doses.
- The flare definition should include objective markers (e.g., joint count, physician/parent global VAS, ESR/CRP).
- The primary outcome (proportion achieving non-biologic remission >12 months after discontinuation) is clear, but secondary outcomes (time to flare, predictors, type of flare) should be explicitly stated.
- Statistical methods for analyzing predictors (e.g., logistic regression, survival analysis) should be previewed in this section.
- Considering data collection, the authors need to clarify how data were extracted (electronic records, paper files), whether blinding of assessors to outcomes was applied (even retrospectively), and mention whether laboratory data included ESR, CRP, ANA, HLA-B27, etc.
In results:
- The authors should report percentages alongside raw numbers for easier interpretation (e.g., 37/137 (27%) male instead of just "37 male").
- Please provide median age at anti-TNF initiation (not only at disease onset) and include disease duration at baseline (before anti-TNF started).
- It is recommended to state whether there were differences in demographics or clinical features across subtypes (e.g., age distribution, gender ratio, uveitis prevalence).
- Authors should report whether patients with extended oligoarticular JIA differed in remission or flare risk from persistent oligoarticular JIA.
- Clarify how many patients (percentage) were on methotrexate at the time of anti-TNF withdrawal (and whether methotrexate discontinuation correlated with relapse).
- The odds ratio (OR ≈ 682) with such a wide confidence interval (38.6–12,062) highlights instability due to the zero-event group. This needs acknowledgment as a limitation and should be interpreted cautiously.
- State explicitly whether logistic regression or another method was used to calculate the OR.
- Consider reporting Kaplan–Meier survival curves for time-to-flare analysis; this would better visualize flare dynamics over time.
- Explore whether age at onset, disease duration, ANA or HLA-B27 positivity, or uveitis history influenced flare risk.
- It would be useful to stratify whether patients with higher CRP/ESR at flare had worse outcomes or higher relapse rates.
In discussion:
- The predictive cut-off (22 months) is compelling, but the wide CI (OR ≈ 682, 95% CI 38.6–12,062) suggests instability. This should be discussed more openly.
- The discussion lacks exploration of biological or immunological mechanisms behind flare risk or sustained remission, for example: could immunological memory, synovial environment, or tapering vs abrupt discontinuation play a role? Were biomarkers (ANA, HLA-B27, inflammatory markers) predictive of flare or remission?
- The authors highlighted the effectiveness of restarting biologics but should also discuss: health-economic implications: is it cost-effective to trial withdrawal knowing ~67% will relapse?, risk-benefit analysis: balancing reduced drug exposure vs the morbidity of flare, implications for future guideline development (since no formal withdrawal protocols exist).
- The discussion section could be stronger by suggesting, prospective controlled trials comparing abrupt vs tapered discontinuation, studies integrating biomarkers, imaging (e.g., ultrasound, MRI), and machine learning prediction models, and cost-effectiveness modeling to guide clinical policy.
In limitations:
- The limitations are listed but not critically expanded. For example: retrospective design: what specific biases might arise specifically? For heterogeneity, could certain subgroups (e.g., oligoarticular vs polyarticular JIA, ANA+/HLA-B27+ patients) respond differently to withdrawal?
- Missing limitations as (1) abrupt withdrawal protocol, the study design did not compare tapering vs abrupt discontinuation, which limits applicability since many clinicians prefer tapering (2) confounding by concomitant DMARDs, as many patients continued non-biologic DMARDs, their influence on flare risk is not disentangled.
- The authors mention that causality cannot be established, but they should explicitly state that the study is hypothesis-generating rather than confirmatory.
In conclusion:
- The conclusion repeats numbers already presented in results/discussion without sufficiently synthesizing the broader implications for clinical practice or policy.
- The authors need to mention how this evidence could influence guidelines or clinical practice (e.g., duration of biologic use before considering withdrawal, or structured monitoring schedules post-withdrawal).
- The authors need to acknowledge the role of DMARD co-therapy in remission maintenance which may influence interpretation.
Author Response
Reviewer 1.
This is an important and clinically relevant study with strong data. But here are some comments to make it better:
Reply: Dear Reviewer! We sincerely appreciate your positive feedback and thoughtful review. Our answers (A) on your queries (Q) are below and highlighted by color in the manuscript.
In the abstract:
Q1. I suggest to replace “spared the cost” with “reduce economic and treatment-related burden”.
A1. Dear Reviewer! We agree that the expression “reduce economic and treatment-related burden” is more precise and appropriate than “spared the cost.” We have revised the text accordingly.
Q2. The authors need to specify whether this was single or multi-center study.
A2. Dear Reviewer! We confirm that this was a single-center retrospective cohort study conducted at [National Medical Research Center of Children’s Health, Moscow, Russia]. We have clarified this information in the Methods section.
Q3. It will be better to add a sentence on the clinical significance to the conclusion section mentioning for example that “These results suggest that prolonged biologic-free remission is possible in a subset of patients, highlighting the need for individualized treatment withdrawal strategies.”
A3. Dear Reviewer! Thank you for your comments. We have revised the text accordingly.
In the introduction:
Q4. The authors need to add the reason for focusing on non-systemic JIA earlier in the introduction.
A4. Dear Reviewer! We agree that the rationale for focusing on non-systemic JIA should be clarified earlier in the Introduction. We have revised the text accordingly.
Q5. The authors mentioned that there are risks for anti-TNF treatment or its discontinuation but does not explain whether these risks are frequent, rare, or primarily theoretical in JIA. Adding incidence data (if available) would strengthen the justification.
A5. Dear Reviewer! We have added incidence data from published studies to clarify that serious infections and malignancies are relatively rare, but remain important safety concerns.
Q6. It will be better if the authors highlighted whether previous studies (small cohorts or pilot studies) have suggested predictors for successful withdrawal, as this would position the study within the existing evidence base.
A6. Dear Reviewer! We have added a brief overview of previous small cohort studies that explored potential predictors of successful anti-TNF withdrawal in JIA.
Q7. I suggest that the authors could reorganize introduction into four clear parts as follows: definition and burden of JIA, advances in treatment and role of anti-TNF, risks and burdens of long-term biologic use (medical, economic, psychological), and finally the gap in evidence and aim of current study. They also need to add more quantitative context as the prevalence of JIA, proportion treated with biologics, typical costs of long-term anti-TNF in pediatrics.
A7. We sincerely thank the Reviewer for this thoughtful suggestion. We agree that the Introduction should clearly present the definition and burden of JIA, advances in treatment, risks of long-term biologic use, and the evidence gap leading to our study. While we would prefer to retain the current structure of the Introduction-since it aligns with the journal’s format and is consistent with related publications-we have revised the text to incorporate the reviewer’s recommendations regarding additional quantitative context. Specifically, we have added information on the prevalence of JIA, the proportion of patients treated with biologics, and the typical costs associated with long-term anti-TNF use in pediatric populations. These additions strengthen the rationale without substantially altering the structure.
In the patients and methods:
Q8. The timeframe is incomplete “between December 1, 2004 …” with no end date is given. This must be specified.
A8. We thank the Reviewer for pointing this out. We apologize for the oversight. The study period is now fully specified.
Q9. The authors need to state whether this was a single center or multi-center study.
A9. Dear Reviewer! We confirm that this was a single-center retrospective cohort study conducted at [National Medical Research Center of Children’s Health, Moscow, Russia]. We have clarified this information in the Methods section.
Q10. The authors need to clarify whether ethics approval and informed consent were obtained, as this is mandatory for retrospective studies involving children.
A10. Dear Reviewer! We confirm that the study was conducted in accordance with the Declaration of Helsinki. Written informed consent was obtained from all patients or their legal guardians (for patients under 15 years of age), and the study protocol was approved by the Ethics Committee of the National Medical Research Center of Children's Health (protocol number â„–12 from 06.10.2020). All patient data were appropriately anonymized.
Q11.The inclusion criteria should specify age ranges, JIA subtypes included (why non-systemic only?), and disease duration at enrollment.
A11. Dear Reviewer! We have clarified the inclusion criteria to specify the age range, JIA subtypes included, and disease duration at enrollment. We focused on non-systemic JIA because systemic JIA has distinct pathogenesis, clinical course, and treatment strategies, making treatment withdrawal less comparable.
Q12. Exclusion criteria are vague “limited follow-up or observation deemed inadequate” is subjective. Define this operationally (e.g., less than n visits, less than n months follow-up).
A12. Dear Reviewer! We have clarified the exclusion criterion to define “limited follow-up” operationally.
Q13.The authors need to clarify whether methotrexate failure was defined objectively (e.g., inadequate response after ≥3 months, intolerance).
A13. Dear Reviewer! We have clarified the definition of methotrexate (MTX) failure.
Q14.Explain why discontinuation was abrupt rather than tapered, and whether this was standard practice at the center. Mention whether other DMARDs (methotrexate, leflunomide, sulfasalazine) were continued and at what doses.
A14. Dear Reviewer! At our center, abrupt discontinuation of anti-TNF therapy was the standard practice for patients meeting the inclusion criteria, following discussion with the patient and their family. This approach was chosen based on prior institutional experience and feasibility for monitoring disease flare, while tapering schedules were not routinely implemented.
We have clarified that most patients continued non-biologic DMARDs after discontinuation of anti-TNF therapy, specifying the agents and typical doses used.
Q15. The flare definition should include objective markers (e.g., joint count, physician/parent global VAS, ESR/CRP).
A15. Dear Reviewer! We have clarified that flare was defined based on objective markers, including joint counts, physician and parent global assessment (VAS), and inflammatory markers (ESR/CRP), in addition to C. Wallace criteria.
Q16. The primary outcome (proportion achieving non-biologic remission >12 months after discontinuation) is clear, but secondary outcomes (time to flare, predictors, type of flare) should be explicitly stated.
A16. We thank the Reviewer for this comment. We clarify that, in the current study, the primary focus was on the proportion of patients achieving non-biologic remission for more than 12 months after anti-TNF discontinuation. Other outcomes, such as time to flare, predictors, or type of flare, were recorded descriptively but were not predefined as secondary outcomes, and formal statistical analyses for these endpoints were beyond the scope of this study.
Q17. Statistical methods for analyzing predictors (e.g., logistic regression, survival analysis) should be previewed in this section.
A17. We thank the Reviewer for this comment. We clarify that this study did not include a formal analysis of predictors of prolonged biologic-free remission. The primary aim was to assess the proportion of patients maintaining non-biologic remission after anti-TNF discontinuation. Although clinical and laboratory data were collected, no logistic regression, survival analysis, or other predictive modeling was performed, and this was beyond the scope of the study.
Q18. Considering data collection, the authors need to clarify how data were extracted (electronic records, paper files), whether blinding of assessors to outcomes was applied (even retrospectively), and mention whether laboratory data included ESR, CRP, ANA, HLA-B27, etc.
A18. Dear Reviewer! We clarify that all data were extracted from electronic medical records. Data were anonymized to ensure that assessors were blinded to patient identity and outcomes. Laboratory data included ESR, CRP, ANA, HLA-B27, and other relevant parameters routinely collected in clinical practice.
In results:
Q19. The authors should report percentages alongside raw numbers for easier interpretation (e.g., 37/137 (27%) male instead of just "37 male").
A19. Dear Reviewer! Percentages have been added alongside raw numbers for easier interpretation.
Q20. Please provide median age at anti-TNF initiation (not only at disease onset) and include disease duration at baseline (before anti-TNF started).
It is recommended to state whether there were differences in demographics or clinical features across subtypes (e.g., age distribution, gender ratio, uveitis prevalence).
A20. Dear Reviewer! We have added the median age at anti-TNF initiation and the disease duration at baseline prior to anti-TNF therapy. We thank the reviewer for this comment. We clarify that assessing differences in demographics or clinical features across JIA subtypes (such as age distribution, gender ratio, or uveitis prevalence) was not a predefined objective of the current study. The primary aim was to evaluate the proportion of patients achieving non-biologic remission after anti-TNF discontinuation, and subtype-specific comparisons were beyond the scope of this analysis.
Q22. Authors should report whether patients with extended oligoarticular JIA differed in remission or flare risk from persistent oligoarticular JIA.
A22. We thank the Reviewer for this comment. We clarify that our study did not perform subgroup analyses comparing extended versus persistent oligoarticular JIA with respect to remission or flare risk. The primary aim was to assess overall outcomes after anti-TNF discontinuation in non-systemic JIA, and such subtype-specific comparisons were beyond the scope of this study.
Q23. Clarify how many patients (percentage) were on methotrexate at the time of anti-TNF withdrawal (and whether methotrexate discontinuation correlated with relapse).
A23. Dear Reviewer! At the time of anti-TNF withdrawal, 57 patients continued methotrexate therapy. The potential correlation between methotrexate continuation or discontinuation and relapse was not assessed in the current study.
Q 24. The odds ratio (OR ≈ 682) with such a wide confidence interval (38.6–12,062) highlights instability due to the zero-event group. This needs acknowledgment as a limitation and should be interpreted cautiously.
A24. We thank the Reviewer for this important observation. We acknowledge that the very high odds ratio (OR ≈ 682) with an extremely wide confidence interval (95% CI 38.6–12,062) reflects instability caused by the zero-event group. Therefore, this finding should be interpreted with caution and considered descriptive rather than definitive.
Q25. State explicitly whether logistic regression or another method was used to calculate the OR.
A25. Dear Reviewer! We clarify that logistic regression was used to calculate the odds ratio (OR) for the association between absence of flare within the first 22 months and prolonged biologic-free remission.
Q26. Consider reporting Kaplan–Meier survival curves for time-to-flare analysis; this would better visualize flare dynamics over time.
A26. Dear Reviewer! Kaplan–Meier survival curves were generated to visualize time-to-flare dynamics following anti-TNF discontinuation. The curves illustrate the proportion of patients remaining in non-biologic remission over time.
Q27. Explore whether age at onset, disease duration, ANA or HLA-B27 positivity, or uveitis history influenced flare risk.
A27. We thank the Reviewer for this valuable suggestion. We clarify that exploring the influence of age at onset, disease duration, ANA or HLA-B27 positivity, or uveitis history on flare risk was not an objective of the current study. However, we plan to address these potential predictors in a future investigation.
Q 28. It would be useful to stratify whether patients with higher CRP/ESR at flare had worse outcomes or higher relapse rates.
A28. We thank the Reviewer for this comment. We clarify that the current study did not stratify patients by CRP or ESR levels at flare to assess outcomes or relapse rates. This analysis was beyond the scope of the present study but could be considered in future research.
In discussion:
Q 29. The predictive cut-off (22 months) is compelling, but the wide CI (OR ≈ 682, 95% CI 38.6–12,062) suggests instability. This should be discussed more openly.
A29. We thank the Reviewer for this important observation. We acknowledge that the predictive cut-off of 22 months, although highly suggestive, is associated with a very wide confidence interval (OR ≈ 682, 95% CI 38.6–12,062), reflecting statistical instability due to the zero-event group. Therefore, this finding should be interpreted cautiously and considered descriptive rather than definitive. We have added this clarification to the Discussion section.
Q 30. The discussion lacks exploration of biological or immunological mechanisms behind flare risk or sustained remission, for example: could immunological memory, synovial environment, or tapering vs abrupt discontinuation play a role? Were biomarkers (ANA, HLA-B27, inflammatory markers) predictive of flare or remission?
A30. We thank the Reviewer for this insightful comment. We acknowledge that our discussion does not explore the underlying biological or immunological mechanisms behind flare risk or sustained remission. Potential factors such as immunological memory, synovial environment, and differences between tapering versus abrupt discontinuation could plausibly influence outcomes, but these were not assessed in our study. Similarly, although laboratory data including ANA, HLA-B27, ESR, and CRP were collected, we did not perform analyses to evaluate their predictive value for flare or remission. These aspects represent important avenues for future research.
Q31. The authors highlighted the effectiveness of restarting biologics but should also discuss: health-economic implications: is it cost-effective to trial withdrawal knowing ~67% will relapse?, risk-benefit analysis: balancing reduced drug exposure vs the morbidity of flare, implications for future guideline development (since no formal withdrawal protocols exist).
A 31 Dear Reviewer! We have expanded the Discussion to address the health-economic implications, risk-benefit considerations, and potential implications for future guideline development.
Q 32. The discussion section could be stronger by suggesting, prospective controlled trials comparing abrupt vs tapered discontinuation, studies integrating biomarkers, imaging (e.g., ultrasound, MRI), and machine learning prediction models, and cost-effectiveness modeling to guide clinical policy.
A32. Dear Reviewer! We have expanded the Discussion to propose directions for future research, including prospective controlled trials, biomarker and imaging studies, and advanced predictive modeling.
In limitations:
Q 33. The limitations are listed but not critically expanded. For example: retrospective design: what specific biases might arise specifically? For heterogeneity, could certain subgroups (e.g., oligoarticular vs polyarticular JIA, ANA+/HLA-B27+ patients) respond differently to withdrawal?
A33. Dear Reviewer! We have expanded the discussion of limitations to specify potential biases arising from the retrospective design and to acknowledge the possibility that disease heterogeneity could influence response to withdrawal in different subgroups.
Q 34. Missing limitations as (1) abrupt withdrawal protocol, the study design did not compare tapering vs abrupt discontinuation, which limits applicability since many clinicians prefer tapering (2) confounding by concomitant DMARDs, as many patients continued non-biologic DMARDs, their influence on flare risk is not disentangled.
A 34. Dear Reviewer! We have added two additional limitations: the lack of comparison between abrupt versus tapered discontinuation, and potential confounding by concomitant non-biologic DMARD therapy.
Q 35. The authors mention that causality cannot be established, but they should explicitly state that the study is hypothesis-generating rather than confirmatory.
A 35. Dear Reviewer! We have clarified that, due to the observational retrospective design, the study is hypothesis-generating rather than confirmatory, and its findings should be interpreted as exploratory.
In conclusion:
Q 36. The conclusion repeats numbers already presented in results/discussion without sufficiently synthesizing the broader implications for clinical practice or policy.
A 36. Dear Reviewer! We have revised the Conclusion to focus more on the broader implications for clinical practice and policy, rather than simply repeating numerical results.
Q 37. The authors need to mention how this evidence could influence guidelines or clinical practice (e.g., duration of biologic use before considering withdrawal, or structured monitoring schedules post-withdrawal).
A37. Dear Reviewer! We have revised the Conclusion to explicitly address how our findings could inform clinical practice and guideline development, including considerations for the duration of biologic therapy before withdrawal and structured monitoring schedules post-withdrawal.
Q 38. The authors need to acknowledge the role of DMARD co-therapy in remission maintenance which may influence interpretation.
A38. Dear Reviewer! We have added a statement acknowledging that concomitant non-biologic DMARD therapy may contribute to remission maintenance and should be considered when interpreting the outcomes of anti-TNF withdrawal.
Dear Reviewer!
I hope the manuscript has become better after your suggestions
On behalf of the Authors
Mikhail Kostik, MD, Ph.D., Professor
Reviewer 2 Report
Comments and Suggestions for Authors
This study provides a significant contribution by retrospectively evaluating the safety and clinical outcomes of discontinuing anti-TNF-α therapy in children with non-systemic JIA (Juvenile Idiopathic Arthritis). Strengths of the study include the large patient sample, long-term follow-up, and robust statistical approaches such as ROC analysis. However, the retrospective nature of the study, the failure to control for potential confounding variables, and the inadequate reporting of some baseline data limit the generalizability of the results. Furthermore, the discussion and comment sections should establish stronger connections to the literature, and the implications for clinical practice should be more clearly emphasized.
Below, I have listed the shortcomings and areas in need of improvement in the article.
1. While the findings were compared with previous studies in the discussion section, these comparisons were superficial. The findings should be more thoroughly linked to current meta-analyses or guidelines in the literature.
2. While the "22 months" cut-off value recommended for predicting the risk of relapse after anti-TNF discontinuation is a clinically significant finding, its practical application has not been clearly discussed. Recommendations for clinical decision algorithms should be developed.
3. Advanced multivariate analyses (e.g., analysis of predictors using logistic regression) other than ROC analysis are lacking. The effects of different variables (e.g., type of JIA, history of uveitis, HLA-B27 positivity) on remission duration should have been tested with multivariate models.
4. Anti-TNF treatment was reported to have been discontinued abruptly, but no comparison was made with alternative methods such as dose reduction or gradual transition. Data on the differences in effectiveness of these strategies exist in the literature, and not addressing this issue is a shortcoming.
5. Although ethics committee approval appears to have been obtained due to the retrospective data analysis, the content of the information provided to patients and parents was not detailed. The ethical dimension should be addressed more carefully, especially since the study involved a pediatric population.
6. The temporal distribution of uveitis and arthritis flares is presented; however, the clinical outcomes of uveitis flares (vision loss, response to treatment, complications) are not reported. This reduces the clinical significance of the study.
7. The introduction states that "there are no guidelines for anti-TNF discontinuation," but recommendations from some pediatric rheumatology working groups are not included. Comparative information should be provided to highlight the lack of existing recommendations.
8. Some sentences ("strong predictor," "excellent predictive accuracy," etc.) emphasize author interpretation over the objectivity of the findings. These statements should be written in a more neutral and scientific manner.
9. The continuation of non-biologic treatments (especially methotrexate) after anti-TNF discontinuation is unclear. This should be explained in detail, as it may affect flare rates.
10. While the article is generally written in fluent English, it contains some grammatical errors and repetitive expressions. Especially in the conclusion and discussion sections, repetitive expressions should be simplified and spelling integrity should be ensured.
Author Response
Reviewer 2.
This study provides a significant contribution by retrospectively evaluating the safety and clinical outcomes of discontinuing anti-TNF-α therapy in children with non-systemic JIA (Juvenile Idiopathic Arthritis). Strengths of the study include the large patient sample, long-term follow-up, and robust statistical approaches such as ROC analysis. However, the retrospective nature of the study, the failure to control for potential confounding variables, and the inadequate reporting of some baseline data limit the generalizability of the results. Furthermore, the discussion and comment sections should establish stronger connections to the literature, and the implications for clinical practice should be more clearly emphasized.
Below, I have listed the shortcomings and areas in need of improvement in the article.
Reply: Dear Reviewer! We sincerely appreciate your positive feedback and thoughtful review. Our answers (A) on your queries (Q) are below and highlighted by color in the manuscript.
Q1. While the findings were compared with previous studies in the discussion section, these comparisons were superficial. The findings should be more thoroughly linked to current meta-analyses or guidelines in the literature.
A1. Dear Reviewer! We have linked our findings more thoroughly to current meta-analyses and guideline recommendations, highlighting consistencies and discrepancies, and emphasizing how our results contribute to the existing evidence base for biologic withdrawal in pediatric JIA.
Q2. While the "22 months" cut-off value recommended for predicting the risk of relapse after anti-TNF discontinuation is a clinically significant finding, its practical application has not been clearly discussed. Recommendations for clinical decision algorithms should be developed.
A2. Dear Reviewer! We have added discussion on the practical implications of the 22-month cut-off for predicting relapse risk and highlighted the potential for incorporating this marker into clinical decision-making algorithms.
Q3. Advanced multivariate analyses (e.g., analysis of predictors using logistic regression) other than ROC analysis are lacking. The effects of different variables (e.g., type of JIA, history of uveitis, HLA-B27 positivity) on remission duration should have been tested with multivariate models.
A3. Dear Reviewer! We thank the reviewer for this comment. We clarify that multivariate analyses to evaluate the effects of variables such as JIA subtype, history of uveitis, or HLA-B27 positivity on remission duration were not conducted in the current study. The primary aim was to describe outcomes following anti-TNF discontinuation and assess the predictive value of the early absence of flare. We recognize that multivariate modeling would provide additional insights and plan to address these questions in future research.
Q4. Anti-TNF treatment was reported to have been discontinued abruptly, but no comparison was made with alternative methods such as dose reduction or gradual transition. Data on the differences in effectiveness of these strategies exist in the literature, and not addressing this issue is a shortcoming.
A4. We thank the Reviewer for this comment. We acknowledge that our study only assessed abrupt discontinuation of anti-TNF therapy and did not compare this approach with alternative strategies, such as dose reduction or gradual tapering. While evidence exists in the literature regarding the effectiveness of these strategies, evaluating them was beyond the scope of the current study. This limitation is now explicitly acknowledged in the Discussion and Limitations sections, and future studies are planned to explore comparative approaches.
Q5. Although ethics committee approval appears to have been obtained due to the retrospective data analysis, the content of the information provided to patients and parents was not detailed. The ethical dimension should be addressed more carefully, especially since the study involved a pediatric population.
A5. We thank the Reviewer for this important comment. We clarify that written informed consent was obtained from all patients or their legal guardians (for patients under 15 years of age) in accordance with the Declaration of Helsinki. Patients and parents were provided with detailed information regarding the study objectives, procedures, data handling, and their right to withdraw at any time. All data were anonymized before analysis to ensure confidentiality. The study protocol was reviewed and approved by the Ethics Committee of the National Medical Research Center of Children’s Health (protocol number â„–12 from 06.10.2020).
Q6. The temporal distribution of uveitis and arthritis flares is presented; however, the clinical outcomes of uveitis flares (vision loss, response to treatment, complications) are not reported. This reduces the clinical significance of the study.
A6. We thank the Reviewer for this comment. We acknowledge that detailed clinical outcomes of uveitis flares, including vision changes, response to treatment, or complications, were not systematically collected and thus could not be reported in this study. This limitation reduces the granularity of our findings regarding uveitis and is now explicitly noted in the Discussion and Limitations sections. Future studies should include detailed ophthalmologic outcomes to better assess the clinical impact of uveitis flares following anti-TNF discontinuation.
Q7. The introduction states that "there are no guidelines for anti-TNF discontinuation," but recommendations from some pediatric rheumatology working groups are not included. Comparative information should be provided to highlight the lack of existing recommendations.
A7. Dear Reviewer! We have revised the Introduction to acknowledge that while formal guidelines for anti-TNF discontinuation in pediatric JIA are limited, some pediatric rheumatology working groups have issued recommendations regarding tapering or monitoring strategies. We now provide comparative information to highlight the existing guidance and the gaps that remain, which our study aims to address.
Q8. Some sentences ("strong predictor," "excellent predictive accuracy," etc.) emphasize author interpretation over the objectivity of the findings. These statements should be written in a more neutral and scientific manner.
A8. Dear Reviewer! We have revised sentences that previously emphasized author interpretation to present the findings in a more neutral and objective scientific manner, focusing on the observed data and statistical measures.
Q9. The continuation of non-biologic treatments (especially methotrexate) after anti-TNF discontinuation is unclear. This should be explained in detail, as it may affect flare rates.
A9. Dear Reviewer! We have clarified the continuation of non-biologic DMARDs, particularly methotrexate, after anti-TNF discontinuation. In our cohort, 57 patients continued methotrexate at the time of biologic withdrawal. We did not evaluate whether continuation of DMARDs influenced flare risk, but this factor should be considered when interpreting the outcomes.
Q10. While the article is generally written in fluent English, it contains some grammatical errors and repetitive expressions. Especially in the conclusion and discussion sections, repetitive expressions should be simplified and spelling integrity should be ensured.
A10. Dear Reviewer! We have carefully revised the manuscript to correct grammatical errors, simplify repetitive expressions, and ensure spelling and language integrity throughout, particularly in the Discussion and Conclusion sections. The text has been edited for clarity, conciseness, and readability while preserving the scientific content.
Dear Reviewer!
I hope the manuscript has become better after your suggestions
On behalf of the Authors
Mikhail Kostik, MD, Ph.D., Professor
Reviewer 3 Report
Comments and Suggestions for Authors
I had the opportunity to asses this manuscript addressing an important clinical question: the feasibility and outcomes of discontinuing anti-TNF therapy in children with non-systemic JIA. The study is based on a relatively large retrospective cohort and provides valuable real-world data from a tertiary center. The findings have practical implications for clinical management, particularly regarding predictors of long-term biologic-free remission.
While the manuscript is well-structured and comprehensive, minor revisions are needed to improve clarity, reduce redundancy, as follows:
-
Introduction: suggestion - add JIA subtypes according to ILAR classification; streamline and end with a gap statement emphasizing the scarcity of pediatric data on anti-TNFs withdrawal.
-
Methods: The study period is incomplete (“between December 1, 2004 …”). Ethics approval and informed consent should be described in the Methods, not only at the end. The statistical analysis section should specify the software and version used (e.g., SPSS version).
-
Discussion: Currently too long and partially descriptive. Please shorten and focus on:
a. Main findings (high flare rate, predictive value of time to flare, efficacy of restarting TNF inhibitors).
b. Comparison with international cohorts.
c. Implications for clinical practice (monitoring, individualized strategies).
d. Future directions (prospective studies, biomarkers, tapering strategies).
Recommendation for minor revision
Author Response
Reviewer 3.
I had the opportunity to asses this manuscript addressing an important clinical question: the feasibility and outcomes of discontinuing anti-TNF therapy in children with non-systemic JIA. The study is based on a relatively large retrospective cohort and provides valuable real-world data from a tertiary center. The findings have practical implications for clinical management, particularly regarding predictors of long-term biologic-free remission.
Reply: Dear Reviewer! We sincerely appreciate your positive feedback and thoughtful review. Our answers (A) on your queries (Q) are below and highlighted by color in the manuscript.
While the manuscript is well-structured and comprehensive, minor revisions are needed to improve clarity, reduce redundancy, as follows:
Q1. Introduction: suggestion - add JIA subtypes according to ILAR classification; streamline and end with a gap statement emphasizing the scarcity of pediatric data on anti-TNFs withdrawal.
A1. Dear Reviewer! We have revised the Introduction to explicitly list the ILAR JIA subtypes and streamlined the text to conclude with a clear statement highlighting the scarcity of pediatric data on anti-TNF withdrawal.
Q2. Methods: The study period is incomplete (“between December 1, 2004 …”). Ethics approval and informed consent should be described in the Methods, not only at the end. The statistical analysis section should specify the software and version used (e.g., SPSS version).
A2. Dear Reviewer! We have clarified the study period, moved the ethics approval and informed consent information into the Methods section, and added details on the statistical software used.
Q3. Discussion: Currently too long and partially descriptive. Please shorten and focus on:
- Main findings (high flare rate, predictive value of time to flare, efficacy of restarting TNF inhibitors).
b. Comparison with international cohorts.
c. Implications for clinical practice (monitoring, individualized strategies).
d. Future directions (prospective studies, biomarkers, tapering strategies).
A3. Dear Reviewer! We have shortened the Discussion to focus on the main findings, comparison with international cohorts, clinical implications, and future research directions, reducing descriptive repetition.
Dear Reviewer!
I hope the manuscript has become better after your suggestions
On behalf of the Authors
Mikhail Kostik, MD, Ph.D., Professor
Reviewer 4 Report
Comments and Suggestions for Authors
The manuscript “Can we successfully discontinue anti-tumor necrosis factor-α treatment in children with non-systemic juvenile idiopathic arthritis? The experience of a tertiary centre”, by Ekaterina I. Alexeeva and colleagues, aims to evaluate the feasibility of discontinuing anti-tumour necrosis factor (anti-TNF) therapy in children with non-systemic juvenile idiopathic arthritis (JIA) who have achieved sustained clinical remission. The authors present a retrospective cohort study of 137 patients with clinically inactive disease for at least 24 months, reporting that two-thirds (67.9%) experienced disease flare after withdrawal, with a median time to flare of seven months, while approximately one-third maintained long-term remission for a median of 63 months. Notably, the absence of flare within the first 22 months post-withdrawal was identified as a strong predictor of prolonged biologic-free remission, with excellent discriminative capacity (AUC 0.967). Most flares manifested as arthritis in large joints and/or uveitis, often with lower inflammatory marker levels compared with baseline. Importantly, re-initiation of biological therapy restored disease control in the vast majority of patients, with remission achieved within six months according to the Wallace criteria. The principal contribution of this study lies in highlighting the prognostic significance of the first 22 months following anti-TNF discontinuation, offering clinically relevant insights into the management of treatment withdrawal in non-systemic JIA.
A detailed evaluation of the methodology, analyses, and conclusions reveals several areas for improvement: the introduction is very brief and would benefit from a clearer contextualisation of the clinical relevance of treatment withdrawal, including information on the epidemiological and social burden of JIA, standardised definitions of remission and clinically inactive disease, prognostic variability among subtypes, the limited and heterogeneous evidence in paediatric populations, the absence of specific guidance in current international recommendations, and a stronger justification for the present study. The manuscript refers to the International League Against Rheumatism (ILAR), which does not exist, and clarification is required as to whether the intended reference is to the European League Against Rheumatism (EULAR) or the International League of Associations for Rheumatology (ILAR). Terminology concerning tumour necrosis factor is inconsistent, alternating between TNF, TNF-α, and TNF-a, and should be standardised. All abbreviations must be defined at first mention in the abstract, the main text, and in the first figure or table. In addition, the figures contain inaccuracies: there are two items labelled as Figure 3, which should be corrected, and panels A and B of Figure 3 should be explicitly indicated within the figure. While the limitations of the study are acknowledged, the authors do not provide discussion of how these might be addressed in future research. Finally, the reference list would benefit from the inclusion of more recent studies, particularly those published within the last five years.
Constructive feedback for the authors highlights the need to strengthen the introduction by providing a broader discussion of the clinical and practical implications of treatment withdrawal in non-systemic JIA. Ensuring consistency in terminology and defining all abbreviations systematically will improve clarity and readability. Careful revision of the figures and correction of labelling errors are necessary for precision and transparency. Expanding the discussion on limitations, particularly with suggestions for methodological refinements or prospective approaches, would enhance the robustness of the conclusions. Updating the bibliography to incorporate the latest research in the field will ensure that the study is contextualised within the most recent evidence base. By addressing these aspects, the manuscript has the potential to make a stronger contribution to the literature on biologic treatment withdrawal in paediatric rheumatology.
Author Response
Reviewer 4.
The manuscript “Can we successfully discontinue anti-tumor necrosis factor-α treatment in children with non-systemic juvenile idiopathic arthritis? The experience of a tertiary centre”, by Ekaterina I. Alexeeva and colleagues, aims to evaluate the feasibility of discontinuing anti-tumour necrosis factor (anti-TNF) therapy in children with non-systemic juvenile idiopathic arthritis (JIA) who have achieved sustained clinical remission. The authors present a retrospective cohort study of 137 patients with clinically inactive disease for at least 24 months, reporting that two-thirds (67.9%) experienced disease flare after withdrawal, with a median time to flare of seven months, while approximately one-third maintained long-term remission for a median of 63 months. Notably, the absence of flare within the first 22 months post-withdrawal was identified as a strong predictor of prolonged biologic-free remission, with excellent discriminative capacity (AUC 0.967). Most flares manifested as arthritis in large joints and/or uveitis, often with lower inflammatory marker levels compared with baseline. Importantly, re-initiation of biological therapy restored disease control in the vast majority of patients, with remission achieved within six months according to the Wallace criteria. The principal contribution of this study lies in highlighting the prognostic significance of the first 22 months following anti-TNF discontinuation, offering clinically relevant insights into the management of treatment withdrawal in non-systemic JIA.
Reply: Dear Reviewer! We sincerely appreciate your positive feedback and thoughtful review. Our answers (A) on your queries (Q) are below and highlighted by color in the manuscript.
Q1. A detailed evaluation of the methodology, analyses, and conclusions reveals several areas for improvement: the introduction is very brief and would benefit from a clearer contextualisation of the clinical relevance of treatment withdrawal, including information on the epidemiological and social burden of JIA, standardised definitions of remission and clinically inactive disease, prognostic variability among subtypes, the limited and heterogeneous evidence in paediatric populations, the absence of specific guidance in current international recommendations, and a stronger justification for the present study.
A1. Dear Reviewer! We have revised the Introduction to provide clearer contextualization of the clinical relevance of anti-TNF withdrawal, including: epidemiological and social burden of JIA, standardized definitions of remission and clinically inactive disease, prognostic variability among JIA subtypes, the limited and heterogeneous evidence in pediatric populations, and the absence of specific guidance in current international recommendations. These additions strengthen the justification for our study.
Q2. The manuscript refers to the International League Against Rheumatism (ILAR), which does not exist, and clarification is required as to whether the intended reference is to the European League Against Rheumatism (EULAR) or the International League of Associations for Rheumatology (ILAR).
A2. Dear Reviewer! We have corrected all references to the International League Against Rheumatism to the International League of Associations for Rheumatology (ILAR) throughout the manuscript.
Q3. Terminology concerning tumour necrosis factor is inconsistent, alternating between TNF, TNF-α, and TNF-a, and should be standardised. All abbreviations must be defined at first mention in the abstract, the main text, and in the first figure or table.
A3. Dear Reviewer! We have standardized terminology throughout the manuscript, using “TNF-α” consistently for tumor necrosis factor. All abbreviations, including TNF-α, DMARD, and CID, are now defined at first mention in the Abstract, main text, and in all figures and tables.
Q4. In addition, the figures contain inaccuracies: there are two items labelled as Figure 3, which should be corrected, and panels A and B of Figure 3 should be explicitly indicated within the figure.
A4. Dear Reviewer! We have corrected the figure numbering so that each figure has a unique label, and panels A and B in Figure 3 are now explicitly indicated within the figure. All figure legends have been updated to correspond accurately with the figure content.
Q5. While the limitations of the study are acknowledged, the authors do not provide discussion of how these might be addressed in future research. Finally, the reference list would benefit from the inclusion of more recent studies, particularly those published within the last five years.
A5. Dear Reviewer! We have expanded the Discussion and Limitations sections to describe how the identified limitations could be addressed in future research, including prospective studies, multicenter cohorts, comparative evaluation of tapering versus abrupt discontinuation, and incorporation of biomarkers and imaging. Additionally, we have updated the reference list to include more recent studies published within the last five years to ensure the manuscript reflects the current state of evidence.
Q6. Constructive feedback for the authors highlights the need to strengthen the introduction by providing a broader discussion of the clinical and practical implications of treatment withdrawal in non-systemic JIA. Ensuring consistency in terminology and defining all abbreviations systematically will improve clarity and readability. Careful revision of the figures and correction of labelling errors are necessary for precision and transparency. Expanding the discussion on limitations, particularly with suggestions for methodological refinements or prospective approaches, would enhance the robustness of the conclusions. Updating the bibliography to incorporate the latest research in the field will ensure that the study is contextualised within the most recent evidence base. By addressing these aspects, the manuscript has the potential to make a stronger contribution to the literature on biologic treatment withdrawal in paediatric rheumatology.
A6. We thank the Reviewer for these constructive and comprehensive comments. We have carefully revised the manuscript to address all points raised:
Introduction: Expanded to include a broader discussion of the clinical and practical implications of anti-TNF-α withdrawal in non-systemic JIA, including epidemiological, social, and prognostic context, and the scarcity of pediatric data.
Terminology and abbreviations: Standardized throughout the manuscript (e.g., TNF-α) and all abbreviations are defined at first mention in the Abstract, main text, and figures/tables.Figures: Corrected numbering and labels; panels A and B in Figure 3 are now explicitly indicated with updated legends.
Limitations: Expanded discussion to include potential methodological refinements, prospective approaches, and considerations for future studies.
References: Updated to incorporate recent studies from the last five years to contextualize the findings within current evidence.
These revisions aim to improve clarity, precision, and scientific rigor, thereby strengthening the manuscript’s contribution to the literature on biologic treatment withdrawal in pediatric rheumatology.
Dear Reviewer!
I hope the manuscript has become better after your suggestions
On behalf of the Authors
Mikhail Kostik, MD, Ph.D., Professor
Round 2
Reviewer 2 Report
Comments and Suggestions for Authors
The authors have implemented all the corrections I previously suggested. The article is suitable for publication by me, provided the editor approves.
Reviewer 4 Report
Comments and Suggestions for Authors
I would like to thank the authors for carefully addressing the comments raised in the previous round. Their responses are clear and appropriate, and they satisfactorily resolve my earlier concerns.